# Post-Harvest Atmospheric Pressure and Composition Modify the Concentration and Bioaccessibility of *α*- and *β*-Carotene in Carrots and Sweet Potatoes

**DOI:** 10.3390/foods12234262

**Published:** 2023-11-25

**Authors:** Batoul Hamieh, Patrick Borel, Sana Raouche, Laurie Bruzzese, Nabil Adjriou, Charlotte Halimi, Gregory Marconot, Guillian Gillet, Jean-Claude Rostain, Régis Guieu, Charles Desmarchelier

**Affiliations:** 1Center for CardioVascular and Nutrition Research (C2VN), Aix-Marseille Univ, INSERM, INRAE, Faculté de Médecine, 27 Boulevard Jean-Moulin, 13005 Marseille, France; batoul.hamieh@univ-amu.fr (B.H.); patrick.borel@univ-amu.fr (P.B.); sana.raouche@univ-amu.fr (S.R.); laurie.bruzzese@univ-amu.fr (L.B.); nabil.adjriou@univ-amu.fr (N.A.); charlotte.halimi@univ-amu.fr (C.H.); gregory.marconot@univ-amu.fr (G.M.); guillian.gillet@univ-amu.fr (G.G.); jean-claude.rostain@univ-amu.fr (J.-C.R.); regis.guieu@univ-amu.fr (R.G.); 2Institut Universitaire de France (IUF), 75000 Paris, France

**Keywords:** carotenoids, vitamin A, bioavailability, process, food matrix

## Abstract

Provitamin A (proVA) carotenoid synthesis and degradation are strongly influenced by environmental factors, including during post-harvest storage. Hypobaric and hyperbaric storages increase the shelf-life of many crops, but their effects on proVA carotenoids are not known. Our aim was to investigate the effects of modifications of atmospheric pressure and composition on *α*- and *β*-carotene concentration and bioaccessibility during the post-harvest storage of carrots and sweet potatoes. Vegetables were stored for 11–14 days at 20 °C in the dark in chambers with modified pressure and O_2_ concentrations. In carrots, *α*- and *β*-carotene concentrations increased significantly during storage, but compared to the control, they were significantly lower in hyperbaria (−23 and −26%, respectively), whereas they did not differ significantly in hypoxia and hypobaria. In sweet potatoes, *α*- and *β*-carotene concentrations decreased significantly during storage, but neither hypoxia, hypobaria nor hyperbaria led to any significant change compared to the control. There was a significant increase for carrot *α*- and *β*-carotene bioaccessibility in hypobaria and hyperbaria, while there was a significant decrease for sweet potato *β*-carotene bioaccessibility in hypobaria/hypoxia and normobaria/hypoxia (−45% and −65% vs. control, respectively). Atmospheric pressure and composition during the post-harvest storage of carrots and sweet potatoes modified the concentration and bioaccessibility of proVA carotenoids.

## 1. Introduction

The term vitamin A (VA) describes a group of fat-soluble molecules exhibiting the biological activity of retinol. VA is essential to human health and is involved in many metabolic and physiological processes, such as vision, cell differentiation, embryonic development and immunity [1]. In humans, VA can be obtained in two different forms: preformed VA (retinol and its esters), obtained exclusively from animal foods; and proVA carotenoids (mainly *β*-carotene, *α*-carotene and *β*-cryptoxanthin), mostly obtained from vegetables and fruits. VA deficiency remains a major public health problem in many low- and middle-income countries, particularly in Southeast Asia and sub-Saharan Africa [2], affecting 190 million children under the age of 5 [3]. Several strategies are used to prevent and treat VA deficiency, notably periodic high-dose preformed VA supplementation, food fortification or dietary diversification. Periodic high-dose VA supplementation is the most widely practiced approach and it has been shown to effectively decrease morbidity and mortality in children [3]. However, it has only a transient and minor positive effect on serum retinol concentration, contrarily to food-based approaches [4,5]. Several food-based approaches are therefore promoted, such as the use of food sources of preformed VA (e.g., animal products and fortified foods), the use of a wider variety of high proVA carotenoid-containing foods and the use of food preparation methods that enhance carotenoid absorption [5].

The carotenoid content of fruits and vegetables at harvesting varies greatly, both between and within species, due to the strict control of carotenoid biosynthesis and accumulation [6]. Multiple genetic and environmental factors, e.g., light, water, temperature, soil pH and composition, are involved. Additionally, post-harvest cultivation practices can also exert a strong influence on carotenoid content, both at the storage and processing stages [6,7]. Indeed, carotenoid biosynthesis and accumulation can be continued during post-harvest storage, especially in climacteric fruits. This is partly due to the increased production of these antioxidative substances in response to the formation of reactive oxygen species induced by the stress condition [7]. On the other hand, carotenoids can degrade upon plant senescence during storage. However, plant senescence can also lead to a degradation of the cellulose structure of the cell wall and a denaturation of carotenoid–protein complexes, thereby favoring carotenoid release from the food matrix during digestion, i.e., their bioaccessibility. Thus, the study of the influence of post-harvest factors on the net amount of carotenoids that is available for absorption is of paramount relevance in order to maximize carotenoid health benefits. Three main post-harvest factors have been studied for their influence on carotenoid food content (reviewed in [7]): temperature, light and atmosphere composition. For most fruits and vegetables, there appears to be a range of optimal temperatures, whereby low temperatures slow down carotenoid biosynthesis and plant senescence, while high temperatures promote carotenoid degradation and plant senescence. For example, one of the present authors (P.B.) showed that carrots stored at 4, 20 and 30 °C for up to 10 days exhibited the highest *α*- and *β*-carotene contents at 20 °C (senescence was observed at 30 °C) and for longer storage durations [8]. Likewise, Imsic et al. showed that carrots stored at 20 °C exhibited a higher increase in (all-*E*)-*β*-carotene than carrots stored at 4 °C [9]. The composition of the storage gas can also be modified to reduce the respiration rate, i.e., %O_2_ is decreased while %CO_2_ is increased, in order to extend fruit and vegetable shelf-life. Such modifications were shown to limit the degradation of carotenoids, most likely that due to the oxidation of these highly unsaturated molecules (self-oxidation or lipoxygenase-induced oxidation) [7]. For example, Simões et al. showed that the content of *β*-carotene in baby carrots stored for 12 days at 4 °C in a controlled atmosphere containing low (0.02 bar) and moderate (0.05 bar) O_2_ amounts was higher than that in baby carrots stored in a normal atmosphere, i.e., *P*_O2_ = 0.21 bar [10].

Another factor that can be modified during the post-harvest storage of fruits and vegetables is atmospheric pressure, as is the case in hyperbaric storage and hypobaric storage, i.e., low-pressure storage (LPS) or sub-atmospheric storage. Hypobaric storage has been shown to increase the shelf-life of many crops, which is mostly attributed to the decrease in respiration rates induced by the associated lower O_2_ partial pressure [11]. Hypobaric storage also results in lower ethylene concentrations due to constant ventilation, and it has been shown to retard or limit microorganism and insect growth. Hyperbaric storage has been defined as “exposing fruit and vegetable to compressed air in a range lower than 10 bar” [12]. It should not be confused with high-pressure processing, where a pressure between 4000 and 12,000 bar is applied, leading to irreversible damage to cell structure. Hyperbaric storage is mainly used to control microorganisms, but it has also been shown to modify respiration rates, ethylene production and the ripening process, as well as color retention, although in a food-specific manner [11]. There are scarce studies that have investigated the effects of hypo- and hyperbaric storage on the nutritional quality of fruits and vegetables, let alone their proVA carotenoid content.

Therefore, the first objective of the present work was to investigate the effects of a change in atmospheric pressure on proVA carotenoid concentrations in carrots and sweet potatoes, two vegetables commonly consumed in countries with VA deficiency [13,14]. Since any modification in atmospheric pressure has an effect on the partial pressure of gases, the composition of the storage gas was also modified in order to differentiate the effects of atmospheric pressure from those of O_2_ partial pressure on carotenoid concentration. Additionally, post-harvest factors can lead not only to the modification of proVA carotenoid concentration, but also to that of their surrounding food matrix, e.g., degradation of the cellulose structure of the cell wall and denaturation of carotenoid–protein complexes. This could in turn modify their bioavailability by modifying their extraction/liberation from the food matrix and their micellization during digestion. Thus, the bioaccessibility of *α*- and *β*-carotene in carrots and sweet potatoes subjected to atmospheric pressure and composition modifications was also assessed using an in vitro gastrointestinal digestion model.

## 2. Materials and Methods

### 2.1. Chemicals and Enzymes

*α*-carotene (≥95% pure), *β*-carotene (≥95% pure), retinyl acetate (≥98% pure), *α*-amylase from *Bacillus* sp., pepsin from porcine gastric mucosa, pancreatin from porcine pancreas, porcine bile extract and taurocholic acid sodium salt hydrate (≥95% pure) were purchased from Sigma-Aldrich (St Quentin Fallavier, France). Ethanol, *n*-hexane and HPLC-grade methanol, methyl tert-butyl ether, dichloromethane, as well as water, were obtained from Carlo-Erba Reagents (Peypin, France).

### 2.2. Vegetable Sources of ProVA Carotenoids

Carrots (*Daucus carota* subsp. *Sativus*) (Nantes type, yaya variety) were organically grown in Sénas (Bouches-du-Rhône department, France). Sweet potatoes (*Ipomoea batatas*) (Beauregard variety) were organically grown in Cadenet (Vaucluse department, France). Both varieties were chosen for their orange flesh, indicative of high *α*- and/or *β*-carotene content. To minimize the effect of uncontrolled post-harvest factors prior to the experiment, the vegetables were used within 2–3 days of harvest.

### 2.3. Modification of Post-Harvest Atmospheric Pressure and Composition 

Although not an SI unit, bar instead of Pa was used as the pressure unit for the sake of convenience (1 bar = 100,000 Pa). All pressures given are absolute pressures.

All experiments were carried out at the Faculty of Pharmacy of Aix-Marseille University (Marseille, France), which is located <100 m above sea level. Two hyperbaric pressure chambers were used, namely a 35 L chamber (maximal pressure 201 bar; Gensollen, Les Pennes Mirabeau, France) and a 100 L chamber (maximal pressure 6 bar; Bethlehem, Bethlehem, PA, USA). Both chambers included analogic pressure manometers and digital pressure analysers (Gefran, Saint-Priest, France). The O_2_ concentration was measured outside the chambers by an oximeter in percentage (OA 135, Servomex, Paris, France) and inside the chambers to control O_2_ partial pressure (Toptronic oxygen monitors, Milan, Italy). The regulation of pressure, O_2_ partial pressure and temperature were performed using a computerized monitoring system (Gefran). Two hypobaric chambers were used, namely a 20 L chamber (hypobaria/hypoxia) and a 250 L chamber (normobaria/hypoxia). The O_2_ concentration inside the chambers was measured with an oxygen sensor (R17 MED, Teledyne Analytical Instruments, Chestnut Street, CA, USA), and the regulation of O_2_ partial pressure was performed by a microcontroller (RP2040, Raspberry Pi, Cambridge, UK) with a 2 × 16 LED screen. For the 250 L chamber, a solenoid valve (RS component, Beauvais, France) was used for N_2_ injection in order to decrease O_2_ partial pressure. For the 20 L chamber, the decrease in pressure was performed by a vacuum pump (VP80, VWR International, Radnor, PA, USA).

Carrots were first gently washed with tap water to remove soil, then dried and separated into sets of 5 with a similar size distribution. All sets were then placed for 2 weeks, a duration chosen to avoid rotting [9], into environments with the atmospheric pressure and composition described in Table 1. The first group of 10 sets was placed under normal atmospheric conditions, i.e., *P* = 1 bar and *P*_O2_ = 0.21 bar, and served as the control group. The second group of 10 sets was placed at *P* = 0.2 bar, an atmospheric pressure found at an altitude of approximately 12,000 m above sea level. Since any modification of atmospheric pressure has an effect on the partial pressure of gases, *P*_O2_ was 0.042 bar. Therefore, this group is henceforth referred to as hypobaria/hypoxia. In order to differentiate the effects of atmospheric pressure from those of O_2_ partial pressure on carotenoid concentration and bioaccessibility, the third group of 10 sets was placed at *P* = 1 bar, and *P*_O2_ was adjusted to 0.03 bar, i.e., close to the *P*_O2_ of the second group, by flushing N_2_. This group is henceforth referred to as normobaria/hypoxia. The fourth group of 10 sets was placed at *P* = 5 bar and O_2_ was fully removed by flushing N_2_. This group is henceforth referred to as hyperbaria/anoxia. The control group and the different chambers were maintained in the dark at 20 °C and water was added in a Petri dish to avoid desiccation. Throughout the incubation period, atmospheric pressure and O_2_ partial pressure were monitored and regulated. Sampling was carried out by removing a set of 5 carrots from each chamber each day of the week, i.e., 10 times in total. The carrots were then cut into 3 parts of equal length (middle, upper, lower), and the middle part was immediately frozen at −80 °C until further analysis.

The above-described experiment was repeated with sweet potatoes, adding another group placed at *P* = 5 bar and *P*_O2_ = 0.21 bar (Table 1). This group is henceforth referred to as hyperbaria/hyperoxia. However, since sweet potatoes take up more space than carrots, storage duration was limited to 11 days instead of 14 because some of the chambers could not accommodate enough sweet potatoes.

### 2.4. Measurement of Vegetable Dry Weight 

The dry weights of carrots and sweet potatoes before and after storage were measured by placing them inside an oven for 24 h at 105 °C (no further weight loss was observed after 24 h in preliminary experiments).

### 2.5. Measurement of ProVA Carotenoid Bioaccessibility

The bioaccessibility of proVA carotenoids in carrots and sweet potatoes was measured using an in vitro digestion model, as previously described [8,15,16]. Of the raw grated carrots or sweet potatoes, 2 g was mixed with a meal consisting of 6.7 g mashed potatoes, 1.2 g ground beef and 200 µL olive oil (all purchased from a local supermarket). The mixture was then ground in 32 mL of 0.9% NaCl (30 s at 6000 rpm) (T18 basic Ultra-Turrax disperser, IKA, Staufen, Germany). The mixture was homogenized for 10 min at 37 °C in a rotating incubator (190 rpm) (Polytest 20, Thermo Fisher Scientific, Illkirch, France). Then, 2.5 mL of an artificial saliva solution was added, and the mixture was further incubated for 10 min at 37 °C under stirring. The pH was then adjusted to 4 ± 0.02 with 1 M HCl. After the addition of 2 mL of a pepsin solution, the mixture was incubated at 37 °C for 30 min under stirring. The pH was then adjusted to 6 ± 0.02 with a 0.9 M NaHCO_3_ buffer before adding 9 mL of a pancreatin solution and 4 mL of a 10% bile solution. The mixture was further incubated for 30 min at 37 °C using the same stirring. At the end of the digestion, the aqueous phase containing mixed micelles was separated from the food particles via centrifugation (1750× *g* for 1 h 12 min at 10 °C). In order to eliminate nonmicellar particles that were recovered in the aqueous phase, it was passed through a 0.22 µm filter (mixed cellulose esters; Merck-Millipore, Molsheim, France) to obtain the micellar phase. The digestate at the end of the duodenal digestion and the micellar phase were collected and weighed, and the samples were stored at −80 °C until lipid extraction and HPLC analysis.

Bioaccessibility was calculated as the ratio of the amount of the given proVA carotenoid found in the micellar phase relative to that of the given proVA carotenoid found in the digestate at the end of the in vitro digestion.

### 2.6. Dynamic Light Scattering

The intensity-weighted mean hydrodynamic radius (mean of 3 technical replicates) and zeta-potential of the particles in the micellar phase recovered after in vitro digestion were determined via dynamic light scattering, using a Zetasizer Nano Zs (Malvern Instruments, Malvern, UK).

### 2.7. ProVA Carotenoid Extraction

Of the raw vegetables, 2 g was crushed with a knife mill (Grindomix GM 200, Retsch, Eragny, France) under liquid nitrogen for 15 s, then homogenized in 20 mL of distilled water. A volume of 500 μL was taken, to which 500 μL of retinyl acetate (internal standard) solubilized in ethanol was added. Double extraction with hexane was performed (with 2 volumes of hexane per volume of the ethanol–sample mixture). After centrifugation at 1200× *g* for 10 min at 4 °C, the hexane phases were combined and evaporated under N_2_ until a dry film was obtained. The samples were solubilized in 200 μL of methanol–dichloromethane (65:35, *v*/*v*) before HPLC analysis.

### 2.8. ProVA Carotenoid Quantification via HPLC

The HPLC system included an Ultimate U3000 separation module (Thermo Fisher Scientific) with an LPG 3400SD pump, a TCC-3000SD column oven and a WPS-3000/TSL autosampler, followed by a DAD-3000 diode array detector. The chromatographic separation and determination of the carotenoids were performed on a YMC Carotenoid C30 column (250 × 4.6 mm, particle size 5 µm, YMC Europe GmbH, Dinslaken, Germany) and a YMC Carotenoid guard column (10 × 4 mm, particle size 5 µm, YMC Europe GmbH), using a mobile phase consisting of eluent A: methanol, eluent B: methyltert-butyl ether and eluent C: H_2_O [96% A, 2% B, 2% C]. The gradient profile of the mobile phase (A:B:C) was set at 96:2:2 and changed linearly to 18:80:2 in 27 min, and then the mobile phase was changed back to 96:2:2 from 31 to 35 min. The flow rate was 1 mL/min and the column temperature was kept constant at 35 °C. The run time was 20 min for *α*- and *β*-carotene, which were detected at 450 nm, and 5 min for retinyl acetate, which was detected at 325 nm. These compounds were identified based on the retention times and consistent spectra of the pure standards (Appendix A). Quantification was performed using Chromeleon software (version 7.2.10 ES, Thermo Fisher Scientific) by comparing the area of the peaks with the standard reference curves. 

### 2.9. Calculations and Statistical Analyses

All results are given as arithmetic means ± SEM of at least 4 experiments. The differences in *α*- or *β*-carotene concentrations during post-harvest storage were analyzed via a two-way ANOVA using a full factorial design, with condition and time as fixed between-subject factors. The molecule, i.e., *α*- and *β*-carotene, was added as a within-subject factor when comparing *α*- and *β*-carotene concentrations. The differences in *α*- or *β*-carotene bioaccessibility, hydrodynamic radius and zeta-potential of the particles in the micellar phase and *α*- or *β*-carotene available amount for uptake were analyzed via a one-way ANOVA. Departures from normality were assessed using the Q–Q plots of standardized residuals. Homogeneity of variances was tested using Levene’s test. The Tukey–Kramer test was used as a post hoc test for pairwise comparisons, while in the case of heteroscedasticity, Welch’s ANOVA was carried out with the Games–Howell test as a post hoc test. Values of *p* < 0.05 were considered significant. Statistical analyses were performed using GraphPad Prism 9.3.1 (GraphPad Software LLC, San Diego, CA, USA) and SPSS 28 (SPSS Inc., Chicago, IL, USA).

## 3. Results

### 3.1. Effect of Atmospheric Pressure and Composition on α- and β-Carotene Concentration in Carrots

Figure 1 shows *α*-carotene concentration as a function of time in carrots placed under different atmospheric conditions for 2 weeks. First, it should be mentioned that carrots did not exhibit any signs of decay during the first 10 days of storage at 20 °C. At days 13 and 14, some carrots in the control and hypobaria/hypoxia groups exhibited the first signs of decay, except in the middle part, which was used for all analyses. There was a significant effect of time (*p* = 0.001), with an overall increase in *α*-carotene concentration during storage. There was also a significant difference in overall *α*-carotene concentration between the different conditions (estimated marginal means in mg/100 g dry weight: control, 13.16 ± 0.61 ^a^; hypobaria/hypoxia, 12.77 ± 0.60 ^a^; normobaria/hypoxia, 13.15 ± 0.60 ^a^; hyperbaria/anoxia, 10.16 ± 0.60 ^b^, *p* < 0.001). 

Figure 2 shows *β*-carotene concentration as a function of time in carrots placed under different atmospheric conditions for 2 weeks. There was a significant effect of time (*p* = 0.003), with an overall increase in *β*-carotene concentration during storage. *β*-Carotene concentration was higher than that of *α*-carotene (estimated marginal means in mg/100 g dry weight: 19.08 ± 0.48 vs. 12.31 ± 0.30; *p* < 0.001), and there was a significant difference in the overall *β*-carotene concentration between the different conditions (estimated marginal means in mg/100 g dry weight: control, 20.85 ± 0.95 ^a^; hypobaria/hypoxia, 20.14 ± 0.96 ^a^; normobaria/hypoxia, 19.38 ± 0.95 ^a b^; hyperbaria/anoxia, 15.94 ± 0.95 ^b^; *p* = 0.002).

### 3.2. Effect of Atmospheric Pressure and Composition on α- and β-Carotene Concentration in Sweet Potatoes

Figure 3 shows *α*-carotene concentration as a function of time in sweet potatoes placed under different atmospheric conditions for 11 days. Sweet potatoes did not exhibit any signs of decay during the 11 days of storage at 20 °C. There was a significant effect of time (*p* = 0.001), with a decrease in *α*-carotene concentration under all conditions during storage. There was no significant difference in the overall *α*-carotene concentration between the different conditions (estimated marginal means in mg/100 g dry weight: control, 1.22 ± 0.06; hypobaria/hypoxia, 1.28 ± 0.63; normobaria/hypoxia, 1.25 ± 0.59; hyperbaria/anoxia, 1.38 ± 0.062; hyperbaria/hyperoxia, 1.36 ± 0.06; *p* = 0.230).

Figure 4 shows *β*-carotene concentration as a function of time in sweet potatoes placed under different atmospheric conditions for 11 days. There was a significant effect of time (*p* = 0.001), with a decrease in *β*-carotene concentration under all conditions during storage. *β*-Carotene concentration was higher than that of *α*-carotene (estimated marginal means in mg/100 g dry weight: 16.56 ± 0.40 vs. 1.30 ± 0.03; *p* < 0.001), but again, there was no significant difference in the overall *β*-carotene concentration between the different conditions (estimated marginal means in mg/100 g dry weight: control, 17.84 ± 0.87; hypobaria/hypoxia, 16.38 ± 0.93; normobaria/hypoxia, 16.98 ± 0.87; hyperbaria/anoxia, 16.11 ± 0.99; hyperbaria/hyperoxia, 15.44 ± 0.87; *p* = 0.273).

### 3.3. Effect of Atmospheric Pressure and Composition on the Bioaccessibility of α- and β-Carotene in Carrots

Modifications of the post-harvest factors can lead not only to the modification of the concentration of proVA carotenoids, but also to that of their surrounding food matrix. This could in turn modify their bioavailability, by modifying their extraction/liberation efficiency from the food matrix and their micellization during digestion. Thus, the bioaccessibility of *α*- and *β*-carotene in the carrots that were placed under the aforementioned modified atmospheric pressure and composition conditions for 10 days was also assessed employing an in vitro digestion model (Figure 5a,b). This specific storage duration was chosen because it was the maximum duration without any apparent decay exhibited by the carrots. The concentration of *α*- and *β*-carotene at day 10 is shown in Appendix A. The bioaccessibility of *α*- and *β*-carotene in carrots placed in hypobaria/hypoxia and hyperbaria/anoxia was significantly increased compared to that of *α*- and *β*-carotene in carrots placed under all other conditions (*α*-carotene: +78% and +67% vs. control, respectively; *β*-carotene: +90% and +48% vs. control, respectively).

Knowing *α*- and *β*-carotene concentrations and bioaccessibilities at day 10, their quantities available for uptake by enterocytes following digestion was calculated as the product of concentration (expressed in mg/100 g total weight, which is a more relevant unit in nutrition) × bioaccessibility (Figure 6a,b). There was no significant difference between the different storage conditions.

### 3.4. Effect of Atmospheric Pressure and Composition on the Bioaccessibility of α- and β-Carotene in Sweet Potatoes

Figure 7a,b shows the bioaccessibility of *α*- and *β*-carotene, respectively, following in vitro digestions of the sweet potatoes that were placed under the aforementioned modified atmospheric pressure and composition conditions for 10 days. The concentration of *α*- and *β*-carotene at day 10 is shown in Appendix A. The same storage duration as that for the carrots was chosen. The bioaccessibility of *α*-carotene in sweet potatoes placed in normobaria/hypoxia and hypobaria/hypoxia was significantly decreased compared to the control (−70 and −39%, respectively), while that of *α*-carotene in sweet potatoes placed in hyperbaria/anoxia and hyperbaria/hyperoxia was significantly increased compared to the control (81% and 41%, respectively). The bioaccessibility of *β*-carotene in sweet potatoes placed in normobaria/hypoxia and hypobaria/hypoxia was also significantly decreased compared to the control (−65 and −45%, respectively), while there was no significant difference in *β*-carotene bioaccessibility between sweet potatoes placed in hyperbaria compared to the control.

Knowing *α*- and *β*-carotene concentrations and bioaccessibilities at day 10, their quantities available for uptake by enterocytes following digestion was calculated as the product of concentration (expressed in mg/100 g total weight, which is a more relevant unit in nutrition) × bioaccessibility (Figure 8a,b). For *α*-carotene, there was a significant difference between the different storage conditions, but the quantities were low, and none differed significantly from the control. For *β*-carotene, although the result of the *F*-test was significant, post hoc tests did not reveal any pairwise significant difference.

### 3.5. Size and Zeta-Potential of Micelles from In Vitro Digestions

In order to characterize some of the mechanisms explaining the observed differences in bioaccessibility, the size and the zeta-potential of the particles found in the micellar phases collected after in vitro digestions were measured. In all micellar phases, micelles represented the main population (the mean diameter comprised between 6 and 8 nm). Regardless of whether the micelles came from the digestion of carrots or sweet potatoes, no significant differences in their size (*p* = 0.113 and 0.45, respectively) and in their zeta-potential (*p* = 0.28 and 0.57, respectively) were observed.

## 4. Discussion

The objective of the present study was to assess whether modifications of atmospheric pressure during the post-harvest storage of carrots and sweet potatoes could modify proVA carotenoid concentrations and bioaccessibilities. Since any modification of atmospheric pressure leads to a modification of gas partial pressure, the carrots and sweet potatoes stored under hypobaric conditions were consequently exposed to lower O_2_ concentrations, i.e., hypoxia. In order to isolate atmospheric pressure as the only factor of variation, the carrots and sweet potatoes stored under hyperbaric conditions were also exposed to hypoxia. This choice is also justified from a nutritional quality standpoint, as a reduction in O_2_ concentration usually results in a lower degradation of carotenoids [7,10]. Nevertheless, for sweet potatoes, as a second hyperbaric chamber was available, it was decided to store them in hyperbaric/hyperoxic conditions. This choice was made to replicate the gas composition typically obtained in hyperbaric storage when the gas composition is not manipulated [11]. Finally, carrots and sweets potatoes were also stored under hypoxic conditions alone in order to differentiate the effects of atmospheric pressure modification (hypo- or hyperbaria) from those of decreased O_2_ partial pressure on carotenoid concentrations and bioaccessibilities. Carrots and sweet potatoes were stored at 20 °C due to the following reasons: (i) one of the present authors (P.B.) has demonstrated that carrots stored at this temperature have higher *α*- and *β*-carotene contents compared to those stored at 4 and 30 °C [8]; (ii) Southeast Asia and sub-Saharan Africa, regions relevant to this study, undergo perennially high temperatures; and (iii) maintaining the different chambers at 20 °C was technically feasible. Additionally, both carrots and sweet potatoes were stored in darkness for the following reasons: (i) studies have indicated that light exposure during post-harvest storage does not significantly affect the concentration of *β*-carotene in carrots [17]; and (ii) from a technical standpoint, it is easier to maintain pressure chambers in a dark environment rather than implementing controlled lighting.

The first noteworthy result is that an overall increase in *α*- and *β*-carotene concentrations in carrots was observed during their post-harvest storage at 20 °C, whereas for sweet potatoes, there was an overall decrease in *α*- and *β*-carotene concentrations. Of note, all concentrations were expressed per dry matter weight, so these differences were not due to the differences in water losses. This finding is consistent with previous studies conducted on carrots stored at 20 °C [8,9]. Few studies were carried out on the retention of *β*-carotene during the storage of fresh sweet potato root. Some have pointed at a decrease in *β*-carotene concentration during storage [18], but it has been shown that *β*-carotene retention highly depends on the temperature of storage and on the variety, with some exhibiting increases in *β*-carotene concentration, while others exhibit decreases [19]. *α*- and *β*-carotene concentrations are influenced by the balance between synthesis processes and degradation processes due to oxidation and isomerization, but the present study does not allow us to determine the relative contribution of each process. Nonetheless, it can be concluded that in carrots stored at 20 °C in the dark for 2 weeks, synthesis appears to be greater than degradation. Conversely, in sweet potatoes, degradation is more prominent than synthesis.

The second noteworthy result of our study is that compared to the concentrations of *α*- and *β*-carotene in carrots stored under normobaric/normoxic conditions, those of carrots stored under hypobaric/hypoxic or normobaric/hypoxic conditions for 14 days did not significantly differ, while a significant decrease in carrots stored under hyperbaric/anoxic conditions was observed. The fact that hypoxia alone did not have an effect on *α*- and *β*-carotene concentrations was surprising, since Simões et al. have shown that the *β*-carotene content in baby carrots stored for at 4 °C in controlled atmospheres containing lower O_2_ amounts for 12 days was higher than that in baby carrots stored in a normal atmosphere (decrease compared to day 0: 0.21 bar, −45%; 0.1 bar −29%; 0.05 bar, −15%; 0.02 bar, −20%) [10]. However, there are several differences between this study and our study. First, baby carrots were prepared by peeling the outer layer of the carrot roots, which exposed them to O_2_, resulting in an increase in carotenoid degradation (both directly and indirectly through the activation of lipoxygenase) [20]. The carrots in the present study were not peeled, thereby lowering O_2_ diffusion to the cortex, where most *β*-carotene, and probably *α*-carotene, are stored [21]. Second, the carrots in the present study were stored at 20 °C, a temperature known to promote higher carotenoid synthesis compared to 4 °C [8]. As a result, the increased carotenoid synthesis occurring at 20 °C may have potentially counteracted the impact of hypoxia on carotenoid degradation, which could have been further minimized by the protective nature of the epidermis.

The concentrations of *α*- and *β*-carotene in carrots stored under hypobaric/hypoxic conditions were not different from those in the control carrots. This, coupled with the observation that hypoxia alone did not impact the concentrations of *α*-carotene and *β*-carotene, strongly indicates that hypobaria alone also did not have an effect. Reduced O_2_ partial pressure is typically considered the primary factor attributed to the effects of hypobaric storage on fruits and vegetables. However, lower ethylene concentration, due to the quicker removal from the plant cells thanks to the constant ventilation in the chambers, is also considered an important factor [11]. A higher ethylene concentration is admitted to have negative effects on carrot organoleptic properties [22,23], but its effects on carotenoid concentration in carrots are not known. In peaches and mandarins, it has been shown to promote carotenoid accumulation [24,25]. Since the experiment was not specifically designed to examine the impact of a low ethylene concentration on *α*- and *β*-carotene concentrations in carrots, and since ethylene concentrations were not measured in the chambers, it can only be cautiously hypothesized that ethylene concentration does not appear to play a significant role in influencing *α*- and *β*-carotene concentrations in carrots stored at 20 °C for 14 days.

Finally, the concentrations of *α*- and *β*-carotene in carrots stored under hyperbaric/hypoxic conditions were lower than those in the control carrots. Since hypoxia alone did not cause any modification of *α*- and *β*-carotene concentrations, this suggests that the observed decrease was mainly due to hyperbaria alone, although a synergistic effect of hyperbaria and hypoxia cannot be completely excluded. To our knowledge, there is no published study on the effect of hyperbaric storage on carrots (excluding carrot juice [26]), whether on preservation, organoleptic or nutritional quality. There is only one study that investigated the effect of hyperbaric storage on carotenoid concentration. Goyette et al. investigated the effects of storage at 1, 3, 5, 7 or 9 bar at 13 °C for up to 15 days on tomatoes at the early breaker stage [27]. Since they did not manipulate gas composition, the tomatoes were also exposed to hyperoxia. Hyperbaria led to a decrease in respiration rate, and it maintained the initial firmness of tomatoes for a longer time. While tomatoes in the control group exhibited a marked increase in their lycopene concentration during storage, this increase was smaller at 3 and 5 bar, and absent at 7 and 9 bar. Although the authors claim that lycopene synthesis was reduced by 57%, 71% and 77% when subjected to hyperbaric treatments of 3, 5 and both 7 and 9 bar, respectively, it is important to consider that this reduction could also be attributed to increased degradation. This is because lycopene concentration is determined by the balance between synthesis and degradation processes. Therefore, although using different vegetable parts, namely the root and fruit, investigating different carotenoids and using different O_2_ concentrations, our results agree in showing that hyperbaria leads to a decrease in carotenoid concentrations. Although high pressure has been shown to modify enzyme activity, this is usually effective at much higher pressures than the ones used in the present study, i.e., typically a few dozens of bar to thousands of bar [28]. However, this is also usually measured during much shorter times, typically a few seconds to a few minutes. It cannot be excluded that low hyperbaria, but sustained for a long time as applied in the present study, could have modified the activity of enzymes, whether involved in carotenoid synthesis or degradation.

In the case of sweet potatoes, there was no effect of the modification of atmospheric pressure and composition on *α*- and *β*-carotene concentrations, although a condition with hyperbaria/hyperoxia was added. In particular, contrary to carrots, hyperbaria, together with hypoxia, did not result in a decrease in *α*- and *β*-carotene concentrations compared to the control. There is no clear explanation for this difference; at this point, only hypotheses can be proposed. Carrots and sweet potatoes differ in their carotenoid localization. Indeed, the predominant plastid type containing *α*- and *β*-carotene in carrots is crystalloid chromoplast, whereas it is crystalloid amylochromoplast in sweet potatoes [29]. It is therefore possible that the effect of hyperbaria on *α*- and *β*-carotene concentrations is affected by the plastid type, i.e., *α*- and *β*-carotene concentrations are little or not affected when located in crystalloid amylochromoplasts. Of note, lycopene in red tomatoes is also stored in crystalloid chromoplasts [29]. Another important point to consider is that, in the present study, *α*- and *β*-carotene synthesis during the post-harvest storage of carrots appeared to be greater than degradation, whereas this was the opposite for sweet potatoes. Thus, if hyperbaria was to exert its negative effect primarily on *α*- and *β*-carotene synthesis, e.g., by decreasing enzymatic activity in the *α*- and *β*-carotene synthesis pathways, and if *α*- and *β*-carotene synthesis was low in sweet potatoes, which could explain the higher degradation vs. synthesis observed in sweet potatoes, this could then explain the different response to hyperbaria between carrots and sweet potatoes. Obviously, further research is needed to confirm these hypotheses and to better understand the mechanisms involved.

The quantities of *α*- and *β*-carotene available for conversion into VA in the human body depend on several factors: (i) the consumption of food sources of proVA carotenoids; (ii) their concentrations in these foods at the moment of ingestion, which are strongly influenced by post-harvest cultivation practices during storage and at the processing stage; and (iii) their bioavailability and conversion efficiency to VA. ProVA carotenoid bioavailability and conversion efficiency to VA display an elevated variability, which is due to several factors (reviewed in [30,31,32]). In the case of carotenoids, their in vivo bioavailability is estimated well by measuring their in vitro bioaccessibility, since good correlations between the two were reported [16,33]. The bioaccessibilities of *α*- and *β*-carotene in carrots stored under normobaric/hypoxic conditions did not significantly differ from those of the control carrots, while those of the carrots stored under hypobaric/hypoxic and hyperbaric/anoxic conditions were significantly higher than those of the control carrots. Since hypoxia alone did not impact the bioaccessibilities of *α*- and *β*-carotene, our results strongly suggest that hypobaria and hyperbaria alone were responsible for the observed effect, although a synergistic effect of hyper-/hypobaria and hypoxia cannot be completely excluded. There are very few studies on the effect of post-harvest storage conditions on proVA carotenoid bioaccessibility. For example, one of the present authors (P.B.) has shown that carrots stored for 6 days at 20 °C had higher proVA carotenoid bioaccessibility than carrots stored for 3 days, although no mechanism explaining this difference was put forward. It is possible that hypo- and hyperbaria led to a modification of the food matrix, e.g., degradation of the cellulose structure of the cell wall and/or denaturation of carotenoid–protein complexes, resulting in a higher carotenoid release during digestion. However, further work, e.g., using microscopy techniques to examine cell structure, is needed to confirm these hypotheses and better understand the mechanisms involved.

The results regarding the bioaccessibility of *α*- and *β*-carotene in sweet potatoes were different from those obtained in carrots. For *α*-carotene, there was a decrease in the bioaccessibility under normobaric/hypoxic and hypobaric/hypoxic conditions, while there was an increase under hyperbaric conditions, both in hypoxia and hyperoxia, compared to the control group. Although the increase in hyperbaria agrees with the results obtained for carrots, the decrease under normobaric/hypoxic and hypobaric/hypoxic conditions does not agree with the results obtained for carrots. For *β*-carotene, storage under hyperbaria, both in hypoxia and hyperoxia, did not result in an increase in bioaccessibility, whereas there was a decrease in bioaccessibility under normobaric/hypoxic and hypobaric/hypoxic conditions. Once again, the mechanisms for such differences are not known.

Finally, when considering the amount available for uptake by enterocytes, which is obtained by the product of concentration x bioaccessibility, it appears that neither hypoxia, nor hypobaria nor hyperbaria resulted in a significant difference compared to the control (except for *α*-carotene in sweet potatoes, but the quantities are so low that they are probably not relevant nutrition-wise). Although it is disappointing that none of these treatments led to an increase, it is also fairly important to stress that there was no decrease. Hence, these post-harvest practices do not seem to have any negative impact on carotenoid availability for uptake, which is correlated to carotenoid bioavailability, at least when they are provided by carrots and sweet potatoes. Therefore, the fact that these practices usually allow for an increase in crop shelf-life means that they could allow for consumers to obtain staple foods with proVA carotenoid bioavailable amounts maintained over longer periods. Obviously, this hypothesis needs to be confirmed for longer periods of storage, and the organoleptic quality of the foods stored as such must be evaluated in order to secure consumer acceptance.

## 5. Conclusions

To conclude, compared to normobaric/normoxic storage, hyperbaric storage of carrots at 20 °C in the dark for 14 days leads to a decrease in *α*- and *β*-carotene concentrations, whereas hypoxic and hypobaric storage results in no difference. Compared to normobaria/normoxia, hypoxia, hypobaria or hyperbaria do not modify *α*- and *β*-carotene concentrations in sweet potatoes stored at 20 °C in the dark for 11 days. In carrots, both hypo- and hyperbaria lead to an increase in *α*- and *β*-carotene bioaccessibilities, whereas in sweet potatoes, hypoxia alone and hypobaria/hypoxia lead to a decrease in *β*-carotene bioaccessibility. 

This study has some limitations. It cannot be fully excluded that the results for carrots or sweet potatoes from other varieties or having followed different crop itineraries would be different. This is unfortunately inherent to most research with vegetal foods, which are characterized by large variability in various quality dimensions. Additionally, the respective contribution of carotenoid synthesis vs. degradation to the differences observed in carotenoid concentrations was not assessed.

However, this study is the first to show the effect of atmospheric pressure modification on *α*- and *β*-carotene concentrations and bioaccessibility in carrots and sweet potatoes, two staple foods highly relevant to proVA carotenoid intakes in developing countries. Carrots stored under hypobaric conditions and sweet potatoes stored under hypobaric or hyperbaric conditions do not exhibit a decrease in their *α*- and *β*-carotene content. In addition, the hypobaric storage of carrots post-harvest led to higher *α*- and *β*-carotene bioaccessibilities. Thus, this study contributes to the identification of post-harvest storage strategies that enhance or maintain nutritional quality, particularly with regard to proVA carotenoid content.

## Figures and Tables

**Figure 1 foods-12-04262-f001:**
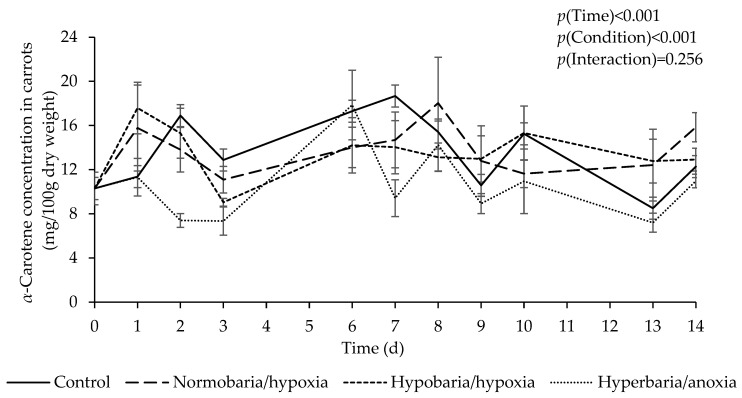
Effect of atmospheric pressure and composition on *α*-carotene concentration in carrots stored at 20 °C for 14 days. Control: *P* = 1 bar, *P*_O2_ = 0.21 bar; normobaria/hypoxia: *P* = 1 bar, *P*_O2_ = 0.03 bar; hypobaria/hypoxia: *P* = 0.2 bar, *P*_O2_ = 0.04 bar; hyperbaria/anoxia: *P* = 5 bar, *P*_O2_ = 0 bar. Values are the means with their SEM represented by vertical bars (*n* = 5).

**Figure 2 foods-12-04262-f002:**
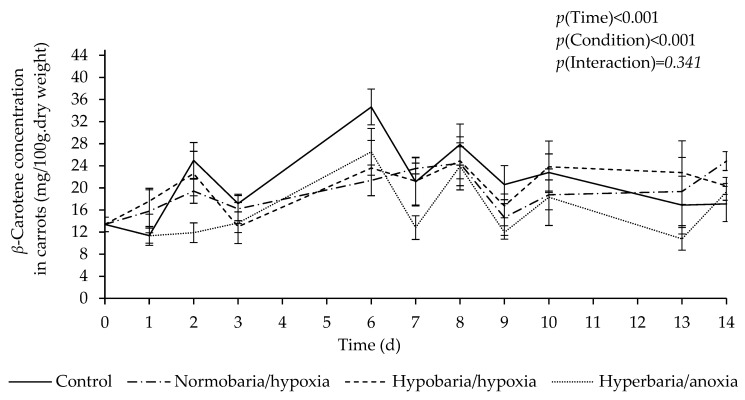
Effect of atmospheric pressure and composition on *β*-carotene concentration in carrots stored at 20 °C for 14 days. Control: *P* = 1 bar, *P*_O2_ = 0.21 bar; normobaria/hypoxia: *P* = 1 bar, *P*_O2_ = 0.03 bar; hypobaria/hypoxia: *P* = 0.2 bar, *P*_O2_ = 0.04 bar; hyperbaria/anoxia: *P* = 5 bar, *P*_O2_ = 0 bar. Values are the means with their SEM represented by vertical bars (*n* = 5).

**Figure 3 foods-12-04262-f003:**
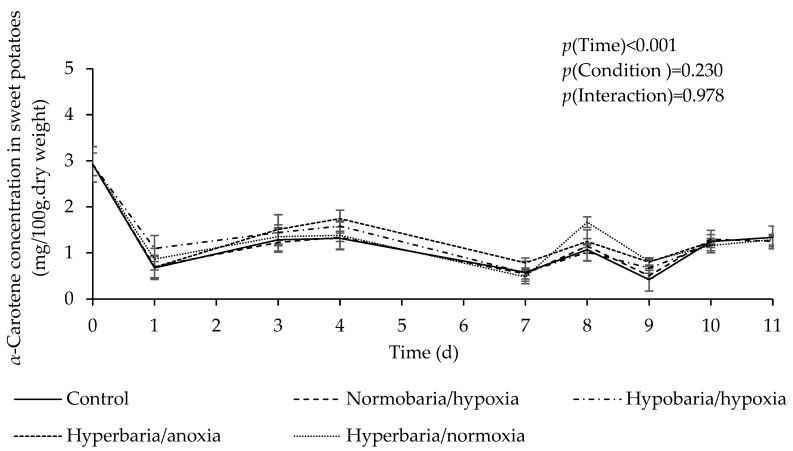
Effect of atmospheric pressure and composition on *α*-carotene concentration in sweet potatoes stored at 20 °C for 11 days. Control: *P* = 1 bar, *P*_O2_ = 0.21 bar; normobaria/hypoxia: *P* = 1 bar, *P*_O2_ = 0.04 bar; hypobaria/hypoxia: *P* = 0.4 bar, *P*_O2_ = 0.02 bar; hyperbaria/anoxia: *P* = 5 bar, *P*_O2_ = 0 bar, hyperbaria/hyperoxia: *P* = 5 bar, *P*_O2_ = 1.05 bar. Values are the means with their SEM represented by vertical bars (*n* = 5).

**Figure 4 foods-12-04262-f004:**
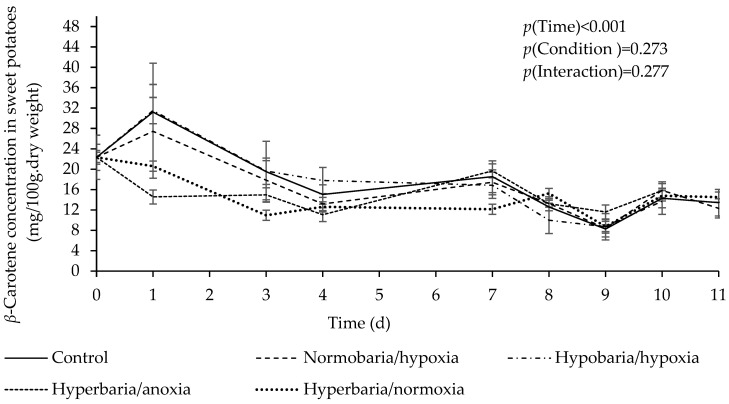
Effect of atmospheric pressure and composition on *β*-carotene concentration in sweet potatoes stored at 20 °C for 11 days. Control: *P* = 1 bar, *P*_O2_ = 0.21 bar; normobaria/hypoxia: *P* = 1 bar, *P*_O2_ = 0.04 bar; hypobaria/hypoxia: *P* = 0.4 bar, *P*_O2_ = 0.02 bar; hyperbaria/anoxia: *P* = 5 bar, *P*_O2_ = 0 bar, hyperbaria/hyperoxia: *P* = 5 bar, *P*_O2_ = 1.05 bar. Values are the means with their SEM represented by vertical bars (*n* = 5).

**Figure 5 foods-12-04262-f005:**
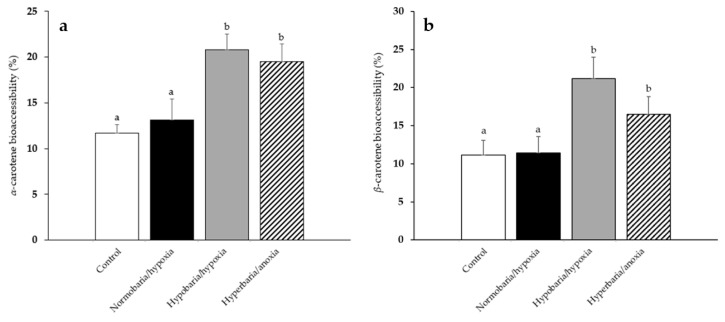
(**a**) Bioaccessibility of *α*-carotene and (**b**) bioaccessibility of *β*-carotene following in vitro digestions of the carrots that were placed under modified atmospheric pressure and composition conditions for 10 days. Control: *P* = 1 bar, *P*_O2_ = 0.21 bar; normobaria/hypoxia: *P* = 1 bar, *P*_O2_ = 0.03 bar; hypobaria/hypoxia: *P* = 0.2 bar, *P*_O2_ = 0.04 bar; hyperbaria/anoxia: *P* = 5 bar, *P*_O2_ = 0 bar. Values are the means with their standard errors represented by vertical bars (*n* = 4). The mean values with unlike superscript letters were significantly different for a given variable (*p* < 0.05; ANOVA followed by the Tukey–Kramer post hoc test).

**Figure 6 foods-12-04262-f006:**
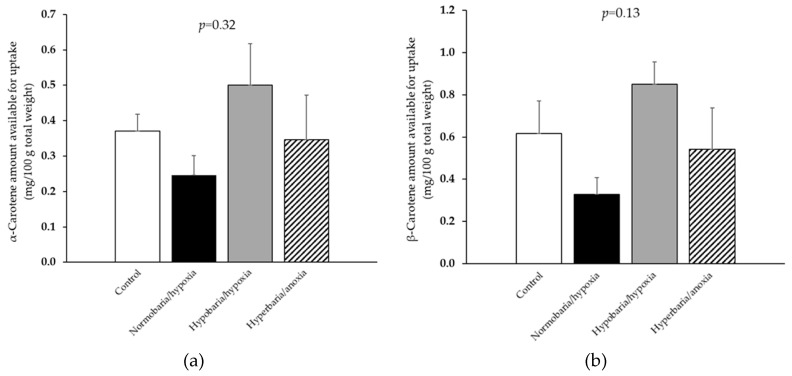
(**a**) Amount of *α*-carotene and (**b**) amount of *β*-carotene available for uptake following in vitro digestions of the carrots that were placed under modified atmospheric pressure and composition conditions for 10 days. Control: *P* = 1 bar, *P*_O2_ = 0.21 bar; normobaria/hypoxia: *P* = 1 bar, *P*_O2_ = 0.03 bar; hypobaria/hypoxia: *P* = 0.2 bar, *P*_O2_ = 0.04 bar; hyperbaria/anoxia: *P* = 5 bar, *P*_O2_ = 0 bar. Values are the means with their standard errors represented by vertical bars (*n* = 4).

**Figure 7 foods-12-04262-f007:**
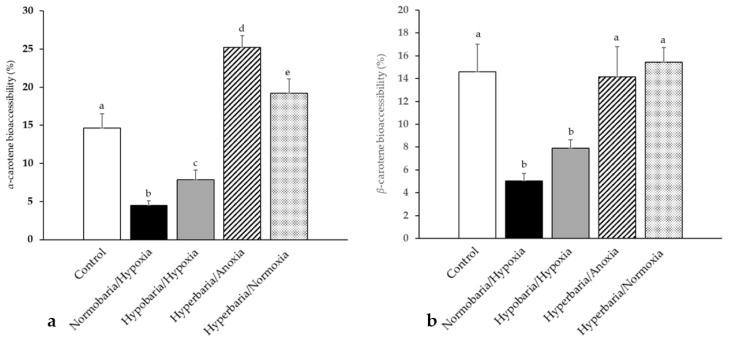
(**a**) Bioaccessibility of *α*-carotene and (**b**) bioaccessibility of *β*-carotene following in vitro digestions of the sweet potatoes that had been placed under modified atmospheric pressure and composition conditions for 10 days. Control: *P* = 1 bar, *P*o_2_ = 0.21 bar; normobaria/hypoxia: *P* = 1 bar, *P*_O2_ = 0.04 bar; hypobaria/hypoxia: *P* = 0.4 bar, *P*_O2_ = 0.02 bar; hyperbaria/anoxia: *P* = 5 bar, *P*_O2_ = 0 bar, hyperbaria/hyperoxia: *P* = 5 bar, *P*_O2_ = 1.05 bar. Values are the means with their standard errors represented by vertical bars (*n* = 4). The mean values with unlike superscript letters were significantly different for a given variable (*p* < 0.05; ANOVA followed by the Tukey–Kramer post hoc test).

**Figure 8 foods-12-04262-f008:**
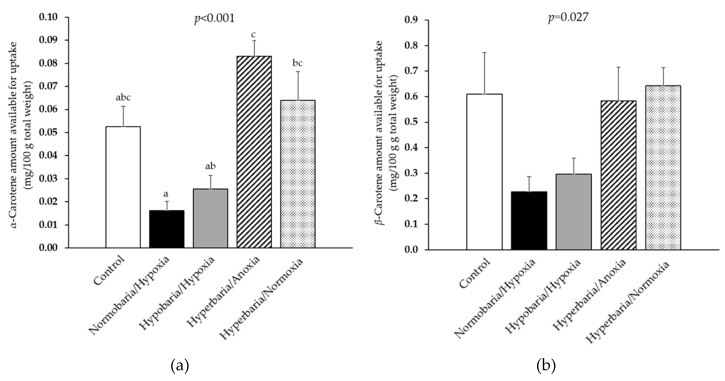
(**a**) Amount of *α*-carotene and (**b**) amount of *β*-carotene available for uptake following in vitro digestions of the sweet potatoes that were placed under modified atmospheric pressure and composition conditions for 10 days. Control: *P* = 1 bar, *P*_O2_ = 0.21 bar; normobaria/hypoxia: *P* = 1 bar, *P*_O2_ = 0.04 bar; hypobaria/hypoxia: *P* = 0.4 bar, *P*_O2_ = 0.02 bar; hyperbaria/anoxia: *P* = 5 bar, *P*_O2_ = 0 bar, hyperbaria/hyperoxia: *P* = 5 bar, *P*_O2_ = 1.05 bar. Values are the means with their standard errors represented by vertical bars (*n* = 4). The mean values with unlike superscript letters were significantly different for a given variable (*p* < 0.05; ANOVA followed by the Tukey–Kramer post hoc test).

**Table 1 foods-12-04262-t001:** Atmospheric pressure and composition conditions.

ProVA Carotenoid Food Source	Condition	Atmospheric Pressure (Bar)	% O_2_	*P*_O2_ (Bar)
Carrots	Control	1	21	0.21
	Normobaria/hypoxia	1	3	0.03
	Hypobaria/hypoxia	0.2	21	0.04
	Hyperbaria/anoxia	5	0	0
Sweet potatoes	Control	1	21	0.21
	Normobaria/hypoxia	1	4	0.04
	Hypobaria/hypoxia	0.4	4	0.02
	Hyperbaria/anoxia	5	0	0
	Hyperbaria/hyperoxia	5	21	1.05

## Data Availability

Data described in the manuscript will be made available upon request to the corresponding author, following application and approval.

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
