# Peer review of "Post-Harvest Atmospheric Pressure and Composition Modify the Concentration and Bioaccessibility of α- and β-Carotene in Carrots and Sweet Potatoes"

_foods, 2023, doi:10.3390/foods12234262_

Round 1

Reviewer 1 Report

Comments and Suggestions for Authors

Very interesting article. Very well described introduction. Well described methodology. The manner of presentation of results commendable. Substantively I have no comments.

My comments:

1/ in the blunt, please articulate:

- the novelty of this work

- research problem

- purpose of the research

- research hypothesis

2/ methodology - for the equipment used, please use the notation: name (model, manufacturer, city, country)

3/ for HPLC analysis - please rearrange the chromatogram containing the analyzed compounds a) standard b) sample

4/ conclusions - it is worth writing them down from the points

Recommendation  a minor revision.

Author Response

Comments and Suggestions for Authors

Very interesting article. Very well described introduction. Well described methodology. The manner of presentation of results commendable. Substantively I have no comments.

We thank the reviewer for the very positive evaluation of our work. We would like to answer the raised points as follows.

My comments:

1/ In the blunt, please articulate:

- the novelty of this work

We have added more details about the novelty of the work in the Conclusion (L587-594 and 601-609).

- research problem

The research problem has been presented in the Introduction (L93-95).

- purpose of the research

The purpose of the research has been presented in the Introduction (L96-109).

- research hypothesis

The research hypothesis is implied by the research problem, before we state the purpose of the research (L93-95). For the sake of readability, we decided not to state it explicitly.

2/ Methodology - for the equipment used, please use the notation: name (model, manufacturer, city, country)

We have added the requested information for every piece of equipment used.

3/ for HPLC analysis - please rearrange the chromatogram containing the analyzed compounds a) standard b) sample

Les chromatogrammes HPLC ont été ajoutés en tant que figure supplémentaire S1.

4/ Conclusions - il vaut la peine de les noter à partir des points

Nous avons suivi ici les recommandations éditoriales de Foods. Néanmoins, nous avons séparé les 3 sous-sections de la Conclusion en 3 paragraphes.

Reviewer 2 Report

Comments and Suggestions for Authors

The aim of the present study was to analyse whether modifications of atmospheric pressure during the post-harvest storage of carrots and sweet potatoes could affect concentrations and bioaccessibility of α- and β- carotene.

The work done is huge and falls within the scope of the journal. However, the possible technological application is not clear and, this is a great limit.

In particular, the Discussion is full of speculations and repetition of results (ethylene was not determined; microscopy was not performed). There are also some methodological limits:

-        Experiment with sweet potatoes lasted 11 days instead of 14 as for carrots: what about the comparison of the two plant samples?

-        Another treatment (Hyperbaria/hyperoxia) was performed for the only sweet potato: and the comparison?

-        Figures 5-8 are related to the treatment of 10 days. What about 14 and 11 days? Maybe a Figure with concentrations of α- and β- carotene at 10 days should be included. Other data should be treated as preliminary results; results on bioaccessibility of carotenes make thus have more sense;

 Others:

-        Line 176: give more details on in vitro digestion model;

-        Line 247: it is not clear. Hyperbaria/anoxia sample did not increase at the end of storage. Maybe another representation with histograms could make clearer the behaviour;

-        Figures 1-4: I cannot see these changes so relevant to justify the importance of an increase in nutraceutical functional properties or a possible technological application of these plant products. The limit of the work is that it is not clear which is the impact from a nutraceutical or technological point of view;

-        Figure 7: about significance of the letters: should increasing letters be related to increasing or decreasing value? In Fig. 5 was the letter b referred to the higher value? Uniform with the other Figures.

Comments on the Quality of English Language

English is fine. Some sentences should be shorter to make them more understandable.

Author Response

The work done is huge and falls within the scope of the journal.

We thank the reviewer for the positive evaluation of our work. We would like to answer the raised points as follows.

However, the possible technological application is not clear and, this is a great limit.

It first should be stressed that this is the first research article to show the effect of atmospheric pressure modification on α- and β-carotene concentrations and bioaccessibility in carrots and sweet potatoes. We had discussed possible technological applications: “Hence, these post-harvest practices do not seem to have any negative impact on carotenoid availability for uptake, which is correlated to carotenoid bioavailability, at least when they are provided by carrots and sweet potatoes. Together with the fact that these practices usually allow for an increase in crop shelf-life therefore means they could allow consumers to obtain for longer periods staple foods whose proVA carotenoid bioavailable amounts are maintained. Obviously, this hypothesis needs to be confirmed for longer periods of storage and the organoleptic quality of the foods thus stored must be evaluated in order to secure consumer acceptance.” (L578-585).

Nevertheless, following Reviewers 2 and 3 comments, we have modified the Conclusion to highlight possible technological applications: “However, this study is the first to show the effect of atmospheric pressure modification on α- and β-carotene concentrations and bioaccessibility in carrots and sweet potatoes, 2 common staple foods highly relevant to proVA carotenoid intakes in developing countries. Carrots stored in hypobaric conditions and sweet potatoes stored in hypobaric or hyperbaric conditions do not exhibit a decrease of their α- and β-carotene content. In addition, hypobaric storage of carrots post-harvest led to higher α- and β-carotene bioaccessibilities. Thus, this study contributes to the identification of post-harvest storage strategies that enhance or maintain nutritional quality, particularly with regard to proVA carotenoid content.” (L601-609).

In particular, the Discussion is full of speculations and repetition of results (ethylene was not determined; microscopy was not performed).

Regarding repetition of results, the mention to ethylene is found only in one paragraph (Discussion, L474-489) whereas there is only one mention to microscopy in the whole manuscript.

There are also some methodological limits:

  • Experiment with sweet potatoes lasted 11 days instead of 14 as for carrots: what about the comparison of the two plant samples?

Indeed, we used different durations for the storage of carrots and sweet potatoes. We explained in the Material and methods the reason for this choice: “since sweet potatoes take up more space than carrots, storage duration was limited to 11 days instead of 14 because some of the chambers could not accommodate enough sweet potatoes.” (L173-175). We did not intend to compare the content and bioaccessibility of provitamin A carotenoids of carrots to those of sweet potatoes during post-harvest storage for several reasons:

  • These vegetables differ in their nutritional quality and their α- and β-carotene contents are also different (very little α-carotene in sweet potatoes as compared to carrots).
  • These vegetables differ in their area of production.
  • These vegetables differ in their culinary use.
  • These vegetables differ in their harvest times.

Hence, to study the effects of hypo- and hyperbaric storage on each vegetable alone is relevant already. Additionally, we think it is more relevant to compare different post-harvest storage strategies within the same food rather than to compare the effect of post-harvest storage strategies between foods that are not mutually substitutable.

  • Another treatment (Hyperbaria/hyperoxia) was performed for the only sweet potato: and the comparison?

Indeed, as we explained (L422-426): “Nevertheless, for sweet potatoes, we had access to a second hyperbaric chamber and we decided to store them in hyperbaric/hyperoxic conditions. This choice was made to replicate the gas composition typically obtained in hyperbaric storage when gas composition is not manipulated [14].”. This condition was always used for comparison with other conditions, i.e. control, normobaria/hypoxia, hypobaria/hypoxia, hyperbaria/anoxia. It cannot be compared to the same condition in carrots since it was only carried out in sweet potatoes. Nevertheless, as we explained in the previous reply, we think it is more relevant to compare different post-harvest storage strategies within the same food.

  • Figures 5-8 are related to the treatment of 10 days. What about 14 and 11 days? Maybe a Figure with concentrations of α- and β- carotene at 10 days should be included. Other data should be treated as preliminary results; results on bioaccessibility of carotenes make thus have more sense;

Indeed, we decided to use carrots and sweet potatoes that had been stored for 10 days, in order to have vegetables without any signs of decay. We justified it in the manuscript as follows: “This specific storage duration was chosen because it was the maximum duration without any apparent decay being exhibited by the carrots.” (L337-339) and “The same storage duration as in carrots was chosen. (L370-371).”.

Following the reviewer’s suggestion, we have added as Supplementary figure S2 and S3 the concentrations α- and β-carotene in carrots and sweet potatoes after 10 days of storage.

Others:

-    Line 176: give more details on in vitro digestion model;

We have added the following: “The mixture was homogenized for 10 min at 37 °C in a rotating incubator (190 rpm) (Polytest 20, Thermo Fisher Scientific, Illkirch, France). Then, 2.5 mL of artificial saliva solution was added, and the mixture was further incubated for 10 min at 37 °C under stirring. The pH was then adjusted to 4 ± 0.02 with 1 M HCl. After the addition of 2 mL of pepsin solution, the mixture was incubated at 37 °C for 30 min under stirring. The pH was then adjusted to 6 ± 0.02 with 0.9 M NaHCO3 buffer before adding 9 mL of a pancreatin solution and 4 mL of a 10% bile solution. The mixture was further incubated for 30 min at 37 °C using the same stirring.”.

-    Line 247: it is not clear. Hyperbaria/anoxia sample did not increase at the end of storage. Maybe another representation with histograms could make clearer the behavior;

     The reviewer is right. This sentence now reads: “There was a significant effect of time (p=0.001), with an overall increase in α-carotene concentration during storage.” (L264-266). Please note that the effect of time is not just measured at the end of the storage time but rather during the whole course of the storage, i.e. 14 days for carrots. This is justified by the fact that there is a fairly high sampling variability, which causes a fairly high inter-day variation. Hence, estimated marginal means, which are provided in the same paragraph, are more informative.

-    Figures 1-4: I cannot see these changes so relevant to justify the importance of an increase in nutraceutical functional properties or a possible technological application of these plant products. The limit of the work is that it is not clear which is the impact from a nutraceutical or technological point of view;

This study deals with actual foods and not with nutraceuticals. Regarding the relevance for possible technological applications, we stress that hypobaric and hyperbaric storage of crops are already in use. They are among several possible strategies for the post-harvest storage of crops. We here show that carrots stored in hypobaric conditions and sweet potatoes stored in hypobaric or hyperbaric conditions do not exhibit a decrease of their α- and β-carotene content. This knowledge is important in order to choose the relevant post-harvest storage strategy. Were these technologies to cause a decrease in the α- and β-carotene content of these vegetables, this would greatly diminish their relevance in developing countries, i.e. where the prevalence of vitamin A deficiency is the highest. Hence, we here show that, as far as provitamin A carotenoid content is concerned, hypobaric post-harvest storage of carrots and hypo- and hyperbaric post-harvest storage of sweet potatoes are relevant strategies. We have changed the Conclusion accordingly: “Carrots stored in hypobaric conditions and sweet potatoes stored in hypobaric or hyperbaric conditions do not exhibit a decrease of their α- and β-carotene content. In addition, hypobaric storage of carrots post-harvest led to higher α- and β-carotene bioaccessibilities. Thus, this study contributes to the identification of post-harvest storage strategies that enhance or maintain nutritional quality, particularly with regard to proVA carotenoid content.” (L604-609).

  • Figure 7: about significance of the letters: should increasing letters be related to increasing or decreasing value? In Fig. 5 was the letter b referred to the higher value? Uniform with the other Figures.

We used standard statistical representation, i.e. as found in textbooks. A different superscript letter indicates a significant, i.e. p<0.05, difference. To know whether this is associated with an increase or decrease, mean values need to be compared. We checked all figures and the y-axis scales used allow readers to see whether a difference between two mean values with different superscript letters is associated with an increase or a decrease.

Commentaires sur la qualité de la langue anglaise

L’anglais, c’est bien. Certaines phrases doivent être plus courtes pour les rendre plus compréhensibles.

Nous avons décomposé les phrases dans la mesure du possible afin de les rendre plus compréhensibles.

Reviewer 3 Report

Comments and Suggestions for Authors

The work is interesting and well organized, however, there are some opportunities for improvement which are mentioned below:

- Avoid using the first person when writing.

in l65 reference 7 is mentioned, improve the wording to clarify what is being referred to.

- Improve the wording in l95

When talking about raw materials, are there any quality characteristics to standardize the experiment (categories, grades, etc.)?

In l130 mention is made of the location of the tests, it would be useful to specify the height above sea level at which they were carried out.

It is necessary to standardise the use of numbers or letters to specify quantities L176 and elsewhere in the document.

In l192 it mentions 3 replicates, in all measurements 3 replicates were made ? review and standardize.

The discussion in general can be improved and a deeper analysis of the results could be done.

The conclusion could include something about possible practical applications of the knowledge generated.

Author Response

The work is interesting and well organized, however, there are some opportunities for improvement which are mentioned below:

We thank the reviewer for the positive evaluation of our work. We would like to answer the raised points as follows.

- Avoid using the first person when writing.

We have removed all occurrences of the first person throughout the whole manuscript.

  • in l65 reference 7 is mentioned, improve the wording to clarify what is being referred to.

This has been corrected. The sentence now reads: “Three main post-harvest factors have been studied for their influence on carotenoid food content (reviewed in [7]): temperature, light and atmosphere composition.” (L64-66).

- Improve the wording in l95

This has been corrected. The sentence now reads: “There are scarce studies that have investigated the effects of hypo- and hyperbaric storage on the nutritional quality of fruits and vegetables, let alone their proVA carotenoid content.” (L93-95).

  • When talking about raw materials, are there any quality characteristics to standardize the experiment (categories, grades, etc.)?

This is a good point. There is of course much variability for many quality characteristics of carrots and sweet potatoes. Additionally, quality is a very multidimensional descriptor, e.g. nutritional quality, hygienic quality, toxicologic quality, organoleptic quality… We specified the variety used and we decided to work with organically grown vegetables because there are very detailed specifications for organic farming and there is thus potentially less variability in crop itineraries than with conventional farming. We also worked with freshly harvested vegetables (2-3 days from field to lab) to decrease uncontrolled post-harvest modifications. Finally, we also reported the concentrations of α- and β-carotene, which can be used as a comparator in other studies. Plant-based foods exhibit significant variability in various quality dimensions and their quality can change much with time. Therefore, it is impossible to provide readers with quality characteristics to standardize the experiment. We can only describe as much as possible the vegetables used and the strategies followed to decrease quality variability. However, we cannot fully exclude that other scientists could find different results with carrots or sweet potatoes due to changes in the very many variables involved in the determination of vegetal food characteristics.

We have added the following sentence in the Conclusion to acknowledge this fact: “This study has some limits: it cannot be fully excluded that results with carrots or sweet potatoes from other varieties or having followed different crop itineraries would be different. This is unfortunately inherent to most research with vegetal foods, which are characterized by large variability in various quality dimensions.” (L595-598).

In l130 mention is made of the location of the tests, it would be useful to specify the height above sea level at which they were carried out.

We do not know the precise height but we have modified the sentence as follows: “All experiments were carried out at the Faculty of Pharmacy of Aix-Marseille University (Marseille, France), which is located <100 m above sea level.” (L131-132).

It is necessary to standardise the use of numbers or letters to specify quantities L176 and elsewhere in the document.

We are here following recommendations found in most style guides for biomedical sciences (e.g., American Medical Association Manual of Style: A Guide for Authors and Editors), i.e. not to start a sentence with a numeral. Where appropriate, we have replaced spelled out numbers by numerals.

In l192 it mentions 3 replicates, in all measurements 3 replicates were made ? review and standardize.

These are technical replicates: each intensity weighted mean hydrodynamic radius measurement for a biological replicate was obtained from 3 technical replicates. As we wrote: “All results are given as arithmetic means ± SEM of at least 4 experiments.” (L243). Additionally, we mention the sample size in each figure legend.

The discussion in general can be improved and a deeper analysis of the results could be done. The conclusion could include something about possible practical applications of the knowledge generated.

We had discussed possible technological applications: “Hence, these post-harvest practices do not seem to have any negative impact on carotenoid availability for uptake, which is correlated to carotenoid bioavailability, at least when they are provided by carrots and sweet potatoes. Together with the fact that these practices usually allow for an increase in crop shelf-life therefore means they could allow consumers to obtain for longer periods staple foods whose proVA carotenoid bioavailable amounts are maintained. Obviously, this hypothesis needs to be confirmed for longer periods of storage and the organoleptic quality of the foods thus stored must be evaluated in order to secure consumer acceptance.” (L578-585).

Néanmoins, à la suite de la recommandation de l’examinateur, nous avons modifié la conclusion en conséquence : « Les carottes entreposées dans des conditions hypobares et les patates douces entreposées dans des conditions hypobares ou hyperbares ne présentent pas de diminution de leur teneur en α et en β-carotène. De plus, l’entreposage hypobare des carottes après la récolte a entraîné une augmentation de la bioaccessibilité du α et du β-carotène. Ainsi, cette étude contribue à l’identification de stratégies de stockage post-récolte qui améliorent ou maintiennent la qualité nutritionnelle, notamment en ce qui concerne la teneur en caroténoïdes proVA. (L601 à 609).

Round 2

Reviewer 2 Report

Comments and Suggestions for Authors

The present version deserves now to be published

Comments on the Quality of English Language

English is nice. Few changes are needed.

Reviewer 3 Report

Comments and Suggestions for Authors

Dear Editor

The authors of the paper have responded adequately to most of my observations and comments. The paper has been improved and I therefore recommend that it be accepted in its present form.